# Chiral recognition of neutral guests by chiral naphthotubes with a bis-thiourea endo-functionalized cavity

Song-Meng Wang[1,3], Yan-Fang Wang [1,3], Liping Huang [1], Li-Shuo Zheng[1], Hao Nian[1], Yu-Tao Zheng[1], Huan Yao[2], Wei Jiang [1] ✉, Xiaoping Wang [1] ✉ & Liu-Pan Yang [2] ✉

Developing chiral receptors with an endo-functionalized cavity for chiral recognition is of great significance in the field of molecular recognition. This study presents two pairs of chiral naphthotubes containing a bis-thiourea endo-functionalized cavity. Each chiral naphthotube has two homochiral centers which were fixed adjacent to the thiourea groups, causing the skeleton and thiourea groups to twist enantiomerically through chiral transfer. These chiral naphthotubes are highly effective at enantiomerically recognizing various neutral chiral molecules with an enantioselectivity up to 17.0. Furthermore, the mechanism of the chiral recognition has been revealed to be originated from differences in multiple non-covalent interactions. Various factors, such as the shape of cavities, substituents of guests, flexibility of host and binding modes are demonstrated to contribute to creating differences in the non-covalent interactions. Additionally, the driving force behind enantioselectivity is mainly attributed to enthalpic differences, and enthalpy-entropy compensation has also been observed to influence enantioselectivity.

Chiral recognition[1-3] is important in both biological processes[4-6] and organic synthesis[7-9]. A better comprehension of chiral recognition is essential to create more effective catalytic systems for asymmetric synthesis[10-14], provide new materials with intriguing chiral properties[15,16] and may contribute to a better understanding of the conservation of homochirality in biological molecules[17-21], as well as guide designing of supramolecular chiral structures[22-25]. Over the past five decades, supramolecular chemists have endeavored to develop chiral hosts that can achieve enantioselective molecular recognition[26-41]. Various non-covalent interactions including hydrogen bonding, ionic, ion-dipole, dipole-dipole, van der Waals as well as π-π interaction play a critical role in differentiating chirality and achieving enantioselectivity ($S_e = K_{R(S)}/K_{S(R)}$, up to ~109)[41]. However, the chiral control of receptors and precise transmission of stereochemical information remains challenging, particularly in simplifying the process of enantioseparation[42] and increasing chirality transfer efficiency[43,44].

In recent years, based on the strategy of simultaneous construction we have reported a series of achiral hosts with an endo-functionalized cavity called naphthotubes[45-52]. These hosts share similar cavity features to bioreceptors and can selectively recognize polar groups and molecules. Therefore, we envisioned that chiral endo-functionalized naphthotubes might achieve chiral recognition for chiral guests. Indeed, chiral amide naphthotubes (Fig. 1a) can enantioselective recognize small organic molecules with decent enantioselectivities (up to 2.0)[53]. This is encouraging, but the enantioselectivity is far from satisfactory. The low enantioselectivity may be attributed to two aspects: (1) the chiral centers of the chiral amide naphthotubes are too distant from the hydrogen bonding sites resulting in a limited chirality transfer; (2) the smaller number of

[1]Department of Chemistry, Southern University of Science and Technology, Xueyuan Blvd 1088, Shenzhen 518055, China. [2]School of Pharmaceutical Science, Hengyang Medical School, University of South China, Hengyang, Hunan 421001, China. [3]These authors contributed equally: Song-Meng Wang, Yan-Fang Wang. ✉e-mail: jiangw@sustech.edu.cn; wangxp@sustech.edu.cn; yanglp@usc.edu.cn

inward-directing hydrogen bonding sites cannot provide sufficient stereochemically dependent binding sites.

In bioreceptors, multiple binding sites often come from side-chains or amino acid residues and are located near chiral centers[54–57]. For example, as shown in Fig. 1b, a chiral cyclic dipeptide cyclo-L-Arg-D-Pro can bind into the chiral cavity of a cyclase with matching binding sites[58]. The proximity of chiral centers and recognition sites increases

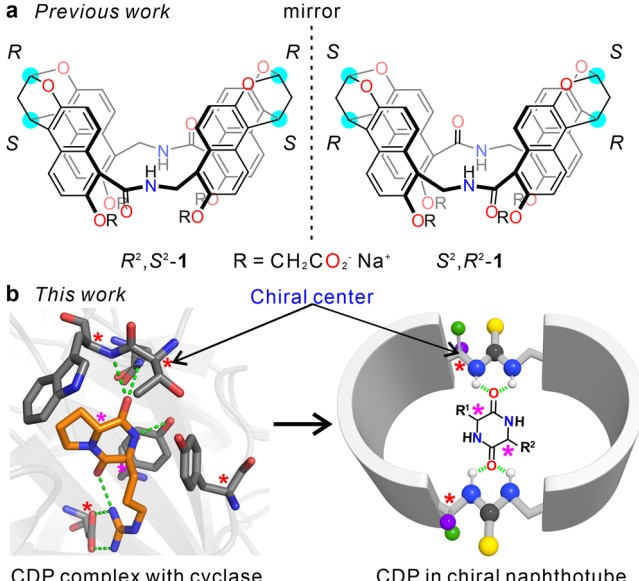

**a** *Previous work*

$R^2,S^2$-**1**　　　R = CH$_2$CO$_2^-$ Na$^+$　　　$S^2,R^2$-**1**

**b** *This work*

Chiral center

CDP complex with cyclase　　　CDP in chiral naphthotube

**Fig. 1 | Design of chiral naphthotubes. a** Chemical structures of chiral amide naphthotubes $R^2,S^2$-**1** and $S^2,R^2$-**1** in previous work, the chiral centers are highlighted with cyan. **b** This work: biomimetic design of an endo-functionalized cavity with chiral centers located at the neighborhood of the inward-directing binding sites. The left figure is a cyclic dipeptide (CDP) cyclo-L-Arg-D-Pro complex with a cyclase (PDB: 5z53), the asterisks indicate the chiral centers.

the efficiency of chiral transfer between the chiral pocket and substrate, thereby contributing a high enantioselectivity in biosystem. Similarly, we wondered if migrating chiral centers closer to hydrogen bonding sites and increasing the number of binding sites could achieve better enantioselectivities. We speculate this strategy may even extend to the enantioselective recognition of bioactive chiral molecules, such as cyclic peptides with chiral centers, as depicted in Fig. 1b. For this purpose, we designed and synthesized two enantiopure bis-thiourea naphthotubes with a chiral center located near each thiourea group (**CT1** and **CT2**, Fig. 2a, b). The chiral naphthotubes exhibit a high level of enantioselectivity (up to 17.0) in recognizing chiral cyclic dipeptides, which is better than the previous reports in terms of enantios-electivity (-9.0)[59]. Furthermore, the chiral naphthotubes can selectively recognize a diverse set of neutral molecules, including cyclic esters, quinuclidinol, oxazolidinones, morpholinone derivatives and drug molecules. Moreover, we have observed an interesting phenomenon in chiral recognition. In general, receptors with $R$ (or $S$) homochiral centers often favor $R$ (or $S$) guests, which is consistent with homochiral selection in nature[60]. However, this is not always the case here, we observed a few instances where $R$ (or $S$) host selected an $S$ (or $R$) guest for the chiral naphthotubes. These cases of heterochiral selectivity encourage further investigation into the underlying molecular mechanisms of enantioselectivity of the chiral naphthotubes.

## Results

### Synthesis and characterization

The key to synthesizing chiral naphthotubes is the production of a chiral diamine with two chiral centers (Supplementary Figs. 1–19). To avoid complicated enantioseparation, a chiral auxiliary *tert*-butane-sulfinamide was employed to regulate the chirality during the addition of a methyl group to the chiral aldimines via Grignard reagent methylmagnesium bromide. For instance, a chiral auxiliary *S*-tert-butanesulfinamide created the chiral carbon center with an $R$ configuration, and vice versa[61,62]. After macrocyclization between diisothio-cyanate and chiral diamine $(R,R)$ under high-dilution conditions, two

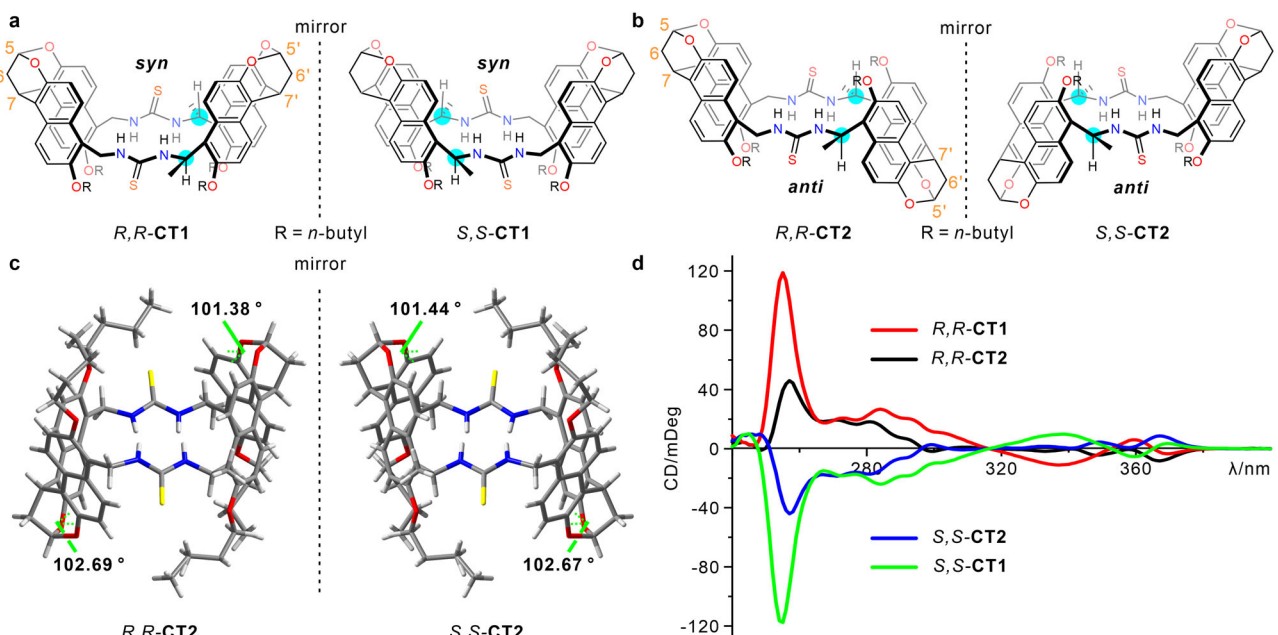

**a**　　　　　mirror

*syn*　　　　　　*syn*

$R,R$-**CT1**　　　R = *n*-butyl　　　$S,S$-**CT1**

**b**　　　　　mirror

*anti*　　　　　　*anti*

$R,R$-**CT2**　　　R = *n*-butyl　　　$S,S$-**CT2**

**c**　　　mirror

101.38 °　　　　101.44 °

102.69 °　　　　102.67 °

$R,R$-**CT2**　　　　$S,S$-**CT2**

**d**

CD/mDeg

— $R,R$-**CT1**
— $R,R$-**CT2**

280　　320　　360　　λ/nm

— $S,S$-**CT2**
— $S,S$-**CT1**

**Fig. 2 | Synthesis and characterization of chiral naphthotubes. a** Chemical structures of chiral bis-thiourea endo-functionalized syn-configured naphthotubes with $R,R$ and $S,S$ chiral centers, and **b** anti-configured naphthotubes with $R,R$ and $S,S$ chiral centers, the chiral centers are highlighted with cyan. **c** X-Ray single crystal structures of $R,R$-**CT2** and $S,S$-**CT2**, the acetone molecules binding in the cavities are removed for clarity, the dihedral angles of bis-naphthalene cleft groups are marked with green lines. **d** Circular dichroism spectra of chiral bis-thiourea endo-functionalized naphthotubes including $R,R$-**CT1**, $R,R$-**CT2**, $S,S$-**CT1**, $S,S$-**CT2** (50 μM in 1,2-dichloroethane, 25 °C).

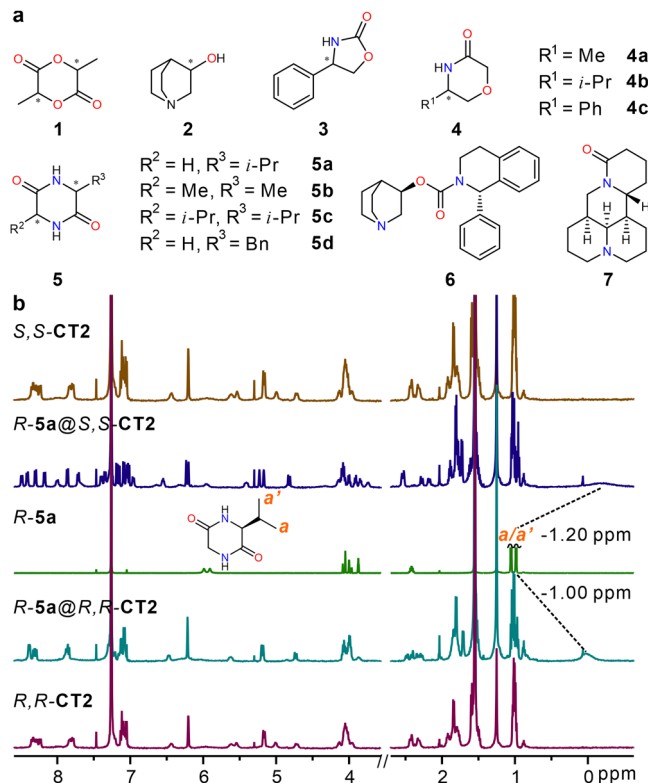

**Fig. 3 | Host-guest interaction. a** Chemical structures of the chiral guests. **b** ¹H NMR spectra (500 MHz, CDCl₃, 0.5 mM, 25 °C) of *R,R*-**CT2**, *R*-**5a**@*R,R*-**CT2**, *R*-**5a**, *R*-**5a**@*S,S*-**CT2** and *S,S*-**CT2** from bottom to top, host-guest complex ratio is 1:1, the change of chemical shift are marked with dash line.

configured isomers including syn-configured naphthotube *R,R*-**CT1** (yield: 5.0%) and anti-configured naphthotube *R,R*-**CT2** (yield: 5.1%) were separated and purified via recrystallization following column chromatography. Two configured isomers are difficult to be assigned according to their complicated NOE signals in the 2D NMR spectra (Supplementary Figs. 20–23). Fortunately, the assignment can be finally confirmed by X-ray single crystal structure of *R,R*-**CT2** (Fig. 2c). Similarly, the other enantiomers *S,S*-**CT1** and *S,S*-**CT2** can be synthesized from the chiral diamine with *S* configuration on two chiral centers. The configured assignments were supported by X-ray single crystal structure of *S,S*-**CT2** (Fig. 2c). In two crystal structures, the skeleton of chiral naphthotubes, especially the thiourea binding sites, undergo enantiomeric twist through intramolecular chiral transfer. Additionally, the asymmetric distribution of chiral centers in the naphthotubes causes two bis-naphthalene clefts to twist with different extents, as evidenced by the structures of X-ray single crystals and energy-minimized simulations. (Fig. 2c, Supplementary Figs. 24–26). We also attempted to introduce more chiral centers to the methylene adjacent to the thiourea groups, but to no avail, likely due to greater steric hindrance. In addition, spectral data also provide evidence of the molecule's twisting, as depicted in Fig. 2d, the enantiomeric pairs of these chiral naphthotubes exhibit mirror-image CD spectra, and offering additional evidence for their enantiotropy of structures and enantiopurity. Chiral high performance liquid chromatography (HPLC) experiments on *R,R*-**CT1**, *R,R*-**CT2**, *S,S*-**CT1** and *S,S*-**CT2** showed that the enantiomeric excess (*ee*) values of these chiral naphthotubes are >99% (Supplementary Fig. 27), further supporting the chirality of the chiral diamines was maintained during macrocyclizations.

## Chiral recognition
Various organic molecules, including cyclic esters (guests **1**), quinuclidinol (guest **2**), oxazolidinones (guests **3**), morpholinones (guests **4**),

cyclic dipeptides (guests **5**) and drug molecules (guests **6** & **7**) were selected to test the recognition with chiral naphthotubes (Fig. 3a). The enantioselectivity was preliminarily evaluated by the ¹H NMR spectra of 1:1 host-guest complex (Supplementary Figs. 28–37). For example as shown in Fig. 3b, a chiral cyclic dipeptide *R*-**5a** interacts with two enantiomeric anti-configured chiral naphthotubes respectively. The differences of chemical shift change of proton **a/a′** between *R*-**5a**@*R,R*-**CT2** and *R*-**5a**@*S,S*-**CT2** demonstrate the enantioselectivity of chiral naphthotubes. These results also provide evidence that enantioselectivity may stem from differences in non-covalent interactions between the host and guest. Moreover, the 1:1 host-guest ¹H NMR experiments in DMSO further confirm the impact of non-covalent interactions on chiral recognition (Supplementary Fig. 38). Additionally, we have attempted to conduct ¹H, ¹H-ROESY experiments of host and guest to obtain evidence of the differences in interactions between enantiomers. (Supplementary Figs. 39–40). However, the nuclear Overhauser effect NOE signal cannot be detected due to the rapid exchange of host-guest exchange on the NMR timescale. The Job plots of *R*-**5a**@*S,S*-**CT2** and high-resolution mass spectra (HRMS) for host-guest complexes support a 1:1 binding stoichiometry (Supplementary Figs. 41–43). Based on the results, the association constants between all possible combinations of hosts and guests were determined by ¹H NMR titration with a 1:1 binding mode (Supplementary Figs. 44–107), and are listed in Table 1. Typically, there are two main methods for calculating the enantioselectivities between a pair of enantiomeric guests and / or hosts. The first method involves utilizing one chiral host that bind to two enantiomeric guests, resulting in different association constants. The second method involves using two enantiomers of hosts, each with different association constants with the same chiral guest. The enantioselectivities obtained from both methods should agree with each other quantitatively and qualitatively. Since the complexes *R*-guest@*R*-host and *R*-guest@*S*-host are the enantiomers of *S*-guest@*S*-host and *S*-guest@*R*-host, respectively, they should have similar stability and thus similar association constants. Therefore, two enantioselectivities should theoretically be similar (or with their reciprocals) and this would support the reliability of the data on the association constants. As shown in Table 1, the enantioselectivities or their reciprocals are all similar for one enantiomeric pair of guests and one enantiomeric pair of hosts. This supports the reliability of our data.

For these chiral guests, two configured chiral naphthotubes both exhibit enantioselectivities, especially in the case of cyclic dipeptides, where the highest enantioselectivity reaches 17.0 for chiral naphthotubes **CT1** and chiral guest *S,S*-**5b**. This is remarkable for a neutral guest molecule and a host with a neutral cavity. Compared with chiral guest *S,S*-**5b**, the chiral guest *S,S*-**1** with similar molecular structure, however, shows a significantly decreased complexation ability and much lower enantioselective recognition. This may be caused by substituent differences in the guest. In addition, the syn-configured naphthotubes **CT1** and the anti-configured naphthotubes **CT2** often show different enantioselectivities to the same chiral guests. Furthermore, the chiral naphthotubes display chiral recognition of guests with bulky substituents and multiple chiral centers, including chiral drugs **6** and **7**. Notably, the chiral naphthotubes exhibit different chiral preferences for selecting chiral guests with homochirality or heterochirality. The homochiral preference exists in most cases, such as **1, 2, 5b, 5c** and **5d** (marked with letter *b* in Table 1), which is consistent with the selection of homochirality in nature. However, in exceptional cases, the chiral naphthotubes exhibit a preference for selecting guests with heterochirality, such as for **3, 4a, 4b, 4c** and **5a** (marked with letter *c* in Table 1). Another noteworthy point is that the syn-configured and anti-configured host exhibit different chiral preferences towards guests **3, 4a, 4b** and **4c**. These findings indicate the shape of the chiral cavities and the variations of substituent groups may potentially influence the chiral recognition and preference. There are still some undiscovered molecular mechanisms behind enantioselectivity exhibited by these

chiral naphthotubes, therefore it is essential to explore the underlying mechanisms.

**Crystal structures and binding mode of chiral recognition**
In order to gain a deeper insight of chiral recognition, we attempted to grow single crystals (Deposition Numbers 2247191 (*R*-**5a**@*S*,*S*-**CT2**), 2247223 (*S*-**5a**@*S*,*S*-**CT2**) contain the supplementary crystallographic data for this paper. These data are provided free of charge by the joint Cambridge Crystallographic Data Centre and Fachinformationszentrum Karlsruhe Access Structures service, the same below) of the host-guest complexes with heterochirality pre-

liminarily. Fortunately, we were successful in obtaining a pair of single crystals of *R*-**5a**@*S*,*S*-**CT2** and *S*-**5a**@*S*,*S*-**CT2**, which were suitable for analysis using X-ray crystallography (Supplementary Figs. 108–109). As shown in Fig. 4a, b, multiple interactions exist between the host and guest, including hydrogen bonding, N-H•••π and C-H•••π interactions. The extra C-H•••π interactions between the isopropyl and naphthyl in *R*-**5a**@*S*,*S*-**CT2** may contribute to the enantioselectivity. Moreover, the different twists of bis-naphthalene clefts create slight asymmetry in the space. This may affect the number and strength of non-covalent interactions. Notably, the enantiomeric guests *S*-**5a** and *R*-**5a** adopt opposite

**Table 1 | The association constants ($K_a$ / M$^{-1}$) and enantioselectivity ($S_e$) of chiral naphthotubes to chiral guests**

|  | *R*,*R*-CT1 | *S*,*S*-CT1 | $S_e$ | *R*,*R*-CT2 | *S*,*S*-CT2 | $S_e$ |
|---|---|---|---|---|---|---|
| *R*,*R*-**1** | $(1.8 \pm 0.1) \times 10^2$ | $(3.7 \pm 0.1) \times 10$ | **4.9**[b] | $(3.5 \pm 0.1) \times 10$ | $(1.8 \pm 0.1) \times 10$ | **1.9**[b] |
| *S*,*S*-**1** | $(4.1 \pm 0.1) \times 10$ | $(1.9 \pm 0.1) \times 10^2$ | **4.6**[-1,b] | $(2.1 \pm 0.1) \times 10$ | $(4.1 \pm 0.1) \times 10$ | **2.0**[-1,b] |
| *R*-**2** | $(1.7 \pm 0.1) \times 10^3$ | $(5.0 \pm 0.1) \times 10^2$ | **3.4**[b] | $(3.6 \pm 0.2) \times 10^2$ | $(3.2 \pm 0.2) \times 10^2$ | **1.1**[b] |
| *R*-**3** | $(2.3 \pm 0.2) \times 10^2$ | $(1.1 \pm 0.1) \times 10^2$ | **2.1**[b] | $(0.9 \pm 0.1) \times 10$ | $(3.6 \pm 0.1) \times 10$ | **4.0**[-1,c] |
| *S*-**3** | $(1.1 \pm 0.1) \times 10^2$ | $(2.3 \pm 0.1) \times 10^2$ | **2.1**[-1,b] | $(3.5 \pm 0.1) \times 10$ | $(1.3 \pm 0.1) \times 10$ | **2.7**[c] |
| *S*-**4a** | $(4.1 \pm 0.3) \times 10^3$ | $(3.2 \pm 0.1) \times 10^3$ | **1.3**[c] | $(9.0 \pm 0.1) \times 10$ | $(6.6 \pm 0.1) \times 10^2$ | **7.3**[-1,b] |
| *S*-**4b** | $(2.8 \pm 0.1) \times 10^3$ | $(3.1 \pm 0.3) \times 10^3$ | **1.1**[-1,b] | $(3.4 \pm 0.5) \times 10^2$ | $(1.8 \pm 0.2) \times 10^2$ | **1.9**[c] |
| *R*-**4c** | $(8.0 \pm 0.4) \times 10^2$ | $(9.0 \pm 0.1) \times 10^2$ | **1.1**[-1,c] | $(2.1 \pm 0.1) \times 10^2$ | $(1.5 \pm 0.1) \times 10^2$ | **1.4**[b] |
| *S*-**4c** | $(1.1 \pm 0.1) \times 10^3$ | $(1.1 \pm 0.2) \times 10^3$ | **1.0** | $(1.7 \pm 0.2) \times 10^2$ | $(1.7 \pm 0.1) \times 10^2$ | **1.0** |
| *R*-**5a** | $(1.0 \pm 0.1) \times 10^4$ | $(5.6 \pm 0.4) \times 10^4$ | **5.6**[-1,c] | $(1.5 \pm 0.2) \times 10^4$ | $(4.6 \pm 0.2) \times 10^4$ | **3.1**[-1,c] |
| *S*-**5a** | $(6.7 \pm 0.3) \times 10^4$ | $(1.3 \pm 0.2) \times 10^4$ | **5.2**[c] | $(4.2 \pm 0.2) \times 10^4$ | $(1.2 \pm 0.1) \times 10^4$ | **3.5**[c] |
| *S*,*S*-**5b** | $(2.3 \pm 0.2) \times 10^3$ | $(3.9 \pm 0.4) \times 10^4$ | **17.0**[-1,b] | $(1.0 \pm 0.1) \times 10^4$ | $(3.0 \pm 0.2) \times 10^4$ | **3.0**[-1,b] |
| *S*,*S*-**5c** | $(8.7 \pm 1.1) \times 10^2$ | $(1.8 \pm 0.1) \times 10^3$ | **2.1**[-1,b] | $(5.0 \pm 0.3) \times 10^2$ | $(5.4 \pm 0.4) \times 10^3$ | **10.8**[-1,b] |
| *S*-**5d** | $(1.6 \pm 0.2) \times 10^4$ | $(2.9 \pm 0.1) \times 10^4$ | **1.8**[-1,b] | $(2.3 \pm 0.3) \times 10^5$ | $(1.4 \pm 0.1) \times 10^{6,a}$ | **6.1**[-1,b] |
| **6** | $(2.2 \pm 0.2) \times 10^3$ | $(1.1 \pm 0.1) \times 10^3$ | **2.0** | $(1.7 \pm 0.1) \times 10^3$ | $(5.4 \pm 0.6) \times 10^2$ | **3.2** |
| **7** | $(3.1 \pm 0.1) \times 10^3$ | $(1.2 \pm 0.1) \times 10^3$ | **2.6** | $(9.1 \pm 0.1) \times 10$ | $(9.9 \pm 0.1) \times 10$ | **1.1**[-1] |

[a]These association constants were determined by competitive NMR titrations; [b]Homochiral selection; [c]Heterochiral selection. All titrations were carried out in toluene-$d_8$ apart from the cyclic dipeptides in CDCl$_3$, and repeated three times for reliability. The enantioselectivities between these enantiomeric hosts and guests were calculated according to the equation of $S_e = K_R / K_S$: for the same chiral guest, $S_e$ is calculated by dividing the association constant of the *R*-configured host by that of the *S*-configured host, and for $S_e$ to be <1, they are written as $(1 / S_e)^{-1}$ for easy comparison.

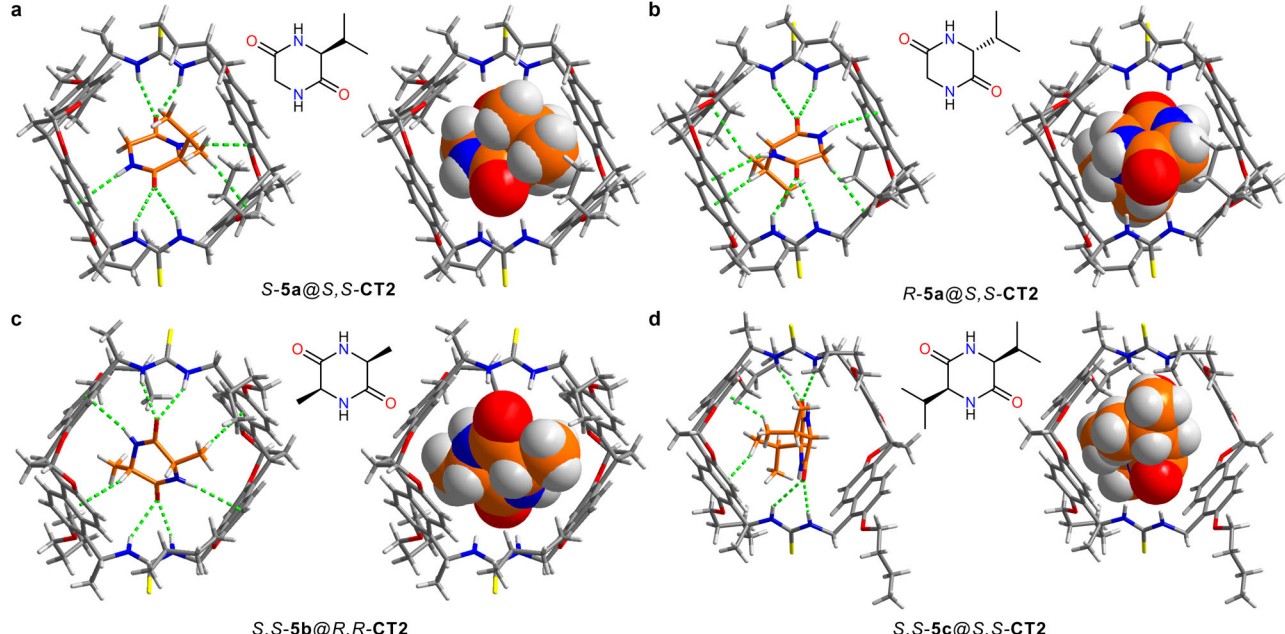

**Fig. 4 | X-Ray single crystal structures of the host-guest complexes. a** *S*-**5a**@*S*,*S*-**CT2**, **b** *R*-**5a**@*S*,*S*-**CT2**, **c** *S*,*S*-**5b**@*R*,*R*-**CT2**, and **d** *S*,*S*-**5c**@*S*,*S*-**CT2**. Green dotted lines indicate noncovalent interaction including hydrogen bonding, NH···π and CH···π interactions, the solvent molecules are removed for clarity.

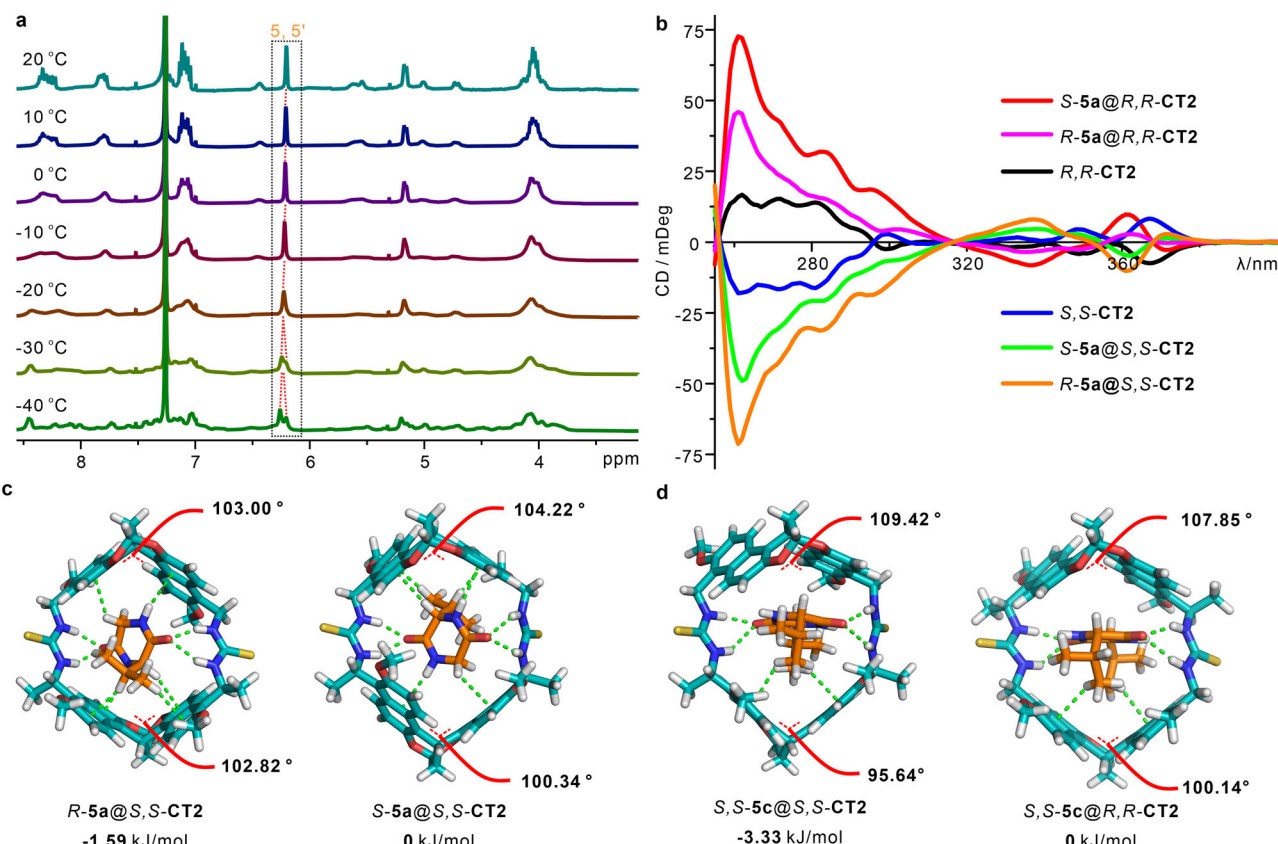

**Fig. 5 | Mechanism investigation of the chiral recognition. a** Variable-temperature [1]H NMR spectra of *S,S*-**CT2** (500 MHz, CDCl₃, 0.5 mM, 25 °C), the temperature decrease from 20 °C to -40 °C. **b** Circular dichroism spectra of anti-configured chiral naphthotubes with *R*-**5a** and *S*-**5a** under saturated binding (25 μM in 1,2-dichloroethane, 25 °C). **c** Energy-minimized structures of *R*-**5a@***S,S*-**CT2,** *S*- 5a@*S,S*-**CT2** and **d** *S,S*-**5c@***S,S*-**CT2,** *S,S*-**5c@***R,R*-**CT2,** which were obtained by DFT (M06-2x/def2-svp) calculations with the PCM solution model in chloroform at 298 K, the relative energies are shown (M06-2x/ma-def2-tzvpp), the dihedral angles of bis-naphthalene clefts in host-guest complexes are marked with red lines, the butoxy are replaced with methoxy to simplify the calculation.

orientations and both lying flat within the cavity of *S,S*-**CT2**. In this binding mode, both enantiomeric guests can snugly fill the twisted cavities of the hosts with structurally complementary space and binding sites.

As for the homochiral selectivity, we tried to obtain single crystals (Deposition Numbers 2247283 (*S,S*-**5c@***S,S*-**CT2**), 2247282 (2**Toluene@***R,R*-**CT2**), 2269211 (*S,S*-**5b@***R,R*-**CT2**)) of *S,S*-**5b**, *S,S*-**5c** and *S,S*-**5d** with anti-configured chiral naphthotubes. However, only the single crystals of *S,S*-**5b@***R,R*-**CT2** and *S,S*-**5c@***S,S*-**CT2** were obtained (Supplementary Figs. 110, 111). Additionally, in the growth of *S,S*-**5c@***R,R*-**CT2** crystal, only two toluene molecules in the cavity of *R,R*-**CT2** were detected in the obtained crystals, despite adding excess *S,S*-**5c** to the solution of *R,R*-**CT2** (Supplementary Fig. 112). The single crystal structures are shown in Figs. 4c, d, multiple interactions stabilize the complexation of the host and guest, and both *S,S*-**5b** and *S,S*-**5c** are observed to stand upright within the cavity, which is distinct from the case of **5a**. Furthermore, considering the opposite binding modes found in case of **5a**, it can be speculated the guests in *S,S*-**5b@***S,S*-**CT2** and *S,S*-**5c@***R,R*-**CT2** should adopt an opposite orientation compared with *S,S*-**5b@***R,R*-**CT2** and *S,S*-**5c@***S,S*-**CT2**, respectively. Overall, the above findings in crystals provided us with evidence of chiral recognition in terms of the differences in non-covalent interactions and opposite binding mode.

### Influence of flexibility on the chiral recognition

The complicated [1]H NMR signals of chiral naphthotubes indicate the potential existence of rapid and asymmetric conformational variation at room temperature. The flexibility of the host may possibly influence chiral recognition. In variable temperature [1]H NMR experiments, as the

temperature decreased, the proton signals of *S,S*-**CT1** and *S,S*-**CT2** split (Fig. 5a, Supplementary Fig. 113) and the signals of protons **5** and **5'** are detected respectively at −40 °C. Additionally, in the [1]H NMR spectra of complex *R*-**5a@***S,S*-**CT2** (Supplementary Fig. 114), the signals of *S,S*-**CT2** split to be clear at room temperature, which likely attribute from the restriction caused by the guest *R*-**5a** in cavity, and there is almost no variation even with a temperature decrease from 20 to −40 °C. In contrast, a larger extent of conformation exchange may exist in *S*-**5a@***S,S*-**CT2** as the temperature decreased (Supplementary Fig. 115), which could be attributed to a potential mismatch in shape between *S*-**5a** and *S,S*-**CT2**.

The circular dichroism (CD) spectra provided additional insights into the conformational changes that occur in chiral naphthotubes upon binding with chiral guests. As depicted in Fig. 5b and Supplementary Figs. 116–117, the mirror-imaged spectra of the enantiomeric pairs suggest that the host-guest complexes undergo a similar twist upon binding. Additionally, the extent of the conformational change is more predominant in host-guest complexes with higher affinity. For example, the CD intensity undergoes a larger extent of enhancement when *S,S*-**CT2** binding with heterochiral guest *R*-**5a** than homochiral guest *S*-**5a**. The results indicate *S,S*-**CT2** may undergo a larger extent of twist when binding with *R*-**5a** than *S*-**5a**, just like wringing a towel with different levels of force. Furthermore, the observed variations in the UV-Vis and fluorescent spectra of the binding process, specifically in relation to the naphthyl groups, provide evidence supporting the involvement of naphthyl groups. On the contrary, when *S,S*-**5b** and *S,S*-**5c** are capsuled into the cavity of anti-configured naphthotubes, respectively. The Cotton signals of *S,S*-**5b@***S,S*-**CT2** and *S,S*-**5c@***S,S*-**CT2** with homochirality show a significant enhancement than those

with heterochirality (Supplementary Figs. 118–119). Besides, the syn-configured chiral naphthotubes also demonstrated a similar result when capturing *S,S*-**5b** into the cavity (Supplementary Fig. 120). The exception is *S*-**5d** was incorporated into *R,R*-**CT2** and *S,S*-**CT2**, respectively, *S*-**5d**@*R,R*-**CT2** exhibited a slightly stronger Cotton signal than *S*-**5d**@*S,S*-**CT2**, possibly due to the additional interactions between the phenyl groups and the host (Supplementary Fig. 121). Above results indicate a conformational variation exists in the binding process, especially for the host-guest enantiomer with higher affinity.

### Theoretical calculations

Based on the binding mode of chiral recognition obtained by single crystals, theoretical calculations were performed using Density Functional Theory (DFT) and considering the solvent effect at 298 K. As shown in Fig. 5c, the stability of *R*-**5a**@*S,S*-**CT2** is higher than that of *S*-**5a**@*S,S*-**CT2**, with a lower free energy of −1.59 kJ/mol, which is generally in qualitative agreement with the $^1$H NMR titration result ($\Delta\Delta G = -2.18$ kJ/mol). And multiple non-covalent interactions, including hydrogen bonding, N-H•••π and C-H•••π interactions, are observed by non-covalent interaction analysis (using independent gradient model based on Hirshfeld partition, IGMH) for both complexes (Supplementary Fig. 122). The results show that the chiral naphthotube *S,S*-**CT2** have stronger non-covalent interactions with *R*-**5a** than *S*-**5a**. These findings demonstrate that the differences of multiple non-covalent interactions should be the basis of the enantioselectivity. Besides, in the complexes of *S,S*-**5c**@*S,S*-**CT2** and *S,S*-**5c**@*R,R*-**CT2**. Stronger non-covalent interactions are also be observed to lead *S,S*-**5c** to prefer to stay in the cavity of *S,S*-**CT2** (Fig. 5d, Supplementary Fig. 123). In like manner, it is similar in the case of *S,S*-**5b** within anti-configured cavities (Supplementary Fig. 124). Through further analysis of the thermodynamic parameters, we found that the driving force behind enantioselectivity is mainly attributed to enthalpic differences, and enthalpy -entropy compensation has also been observed to influence enantioselectivity (Supplementary Tab. 2). And the entropy change may be caused by the different conformational twists in chiral recognition, which are indicated by the dihedral angles of bis-naphthalene clefts (Fig. 5c, d).

The calculations further confirmed the opposite binding mode in the chiral recognition. Therefore, we also carried out the related calculation of *S,S*-**5b** with an opposite orientation binding mode in syn-configured chiral naphthotube. The non-covalent interaction analysis and thermodynamic parameters reveal that the enantioselectivity is mainly attributed to enthalpic differences (Supplementary Fig. 125, Supplementary Tab. 2). From the exploration of the chiral recognition mechanism of cyclic dipeptides mentioned above, it can be seen the enantioselectivity whether homochiral or heterochiral selectivity originate from the differences of multiple non-covalent interactions. And the shape of cavities, substituents of guests, flexibility of host and binding modes, are demonstrated to contribute to creating differences in the non-covalent interactions.

Based on these findings regarding chiral recognition, we proceeded to investigate the chiral recognition of chiral guests **4** within the chiral naphthotubes. Specifically, we focused on three guests, *S*-**4a**, *S*-**4b** and *S*-**4c**, which all share morpholinone groups. These groups contain two different hybridized oxygen atoms that can act as hydrogen-bonding receptors. Thus, the chiral guests **4** can adopt four possible binding modes in an asymmetrical cavity. The results of theoretical calculations indicate the chiral guests **4** adopt a similar opposite binding mode within chiral cavities of naphthotubes (Supplementary Figs. 126–129). Further analysis of spectral and thermodynamic data indicated that the enantioselectivity was mainly driven by enthalpy, resembling the driven force observed in chiral guests **5** (Supplementary Figs. 130–131, Supplementary Tab. 2). These findings are generally in qualitative agreement with the $^1$H NMR and spectral results. Moreover, the results again highlight the importance of multiple non-covalent interactions in chiral recognition.

## Discussion

Reviewing the mechanisms of chiral recognition in a previous study, this mechanism is similar to the four-location model used to explain protein's ability to discriminate between L- and D-isomers in biosystem[63]. In four-location model, a minimum of four designated locations may include a direction needed for chiral discrimination. In the case here, the enantioselectivity was found to originate from variations in multiple non-covalent interactions also include an orientation, thus, which should be referred to as the multipoint location model. The above investigations have provided us with a better understanding of the molecular basis of chiral selection, and additionally to comprehend the conservation of homochirality in biosystems.

In summary, we have presented the synthesis, characterization, chiral recognition, and mechanism of chiral recognition of two pairs of chiral naphthotubes. The thiourea groups with fixed chiral centers are responsible for the molecular skeleton, which includes a twist in an enantiomeric manner. The validity of the chiral introduction method was confirmed by single crystal structures and CD spectra. The presence of multiple endo-functionalized hydrogen bonding sites enhances the ability of the naphthotubes to selectively capture various chiral guests, including cyclic esters, quinuclidinol, oxazolidinones, morpholinone derivatives, cyclic peptides and drug molecules. Furthermore, we used various techniques, including $^1$H NMR titrations, circular dichroism spectroscopy, X-ray single crystallography and DFT calculations reveal the mechanism and the driving force of enantioselectivity. This study provides a comprehensive understanding of the molecular basis of enantioselectivity within the chiral naphthotubes. Moreover, it is helpful to comprehend the conservation of homochirality in nature and guide the designing of chiral supramolecular receptors.

## Methods

### General

All the reagents involved in this research were commercially available and used without further purification unless otherwise noted. Solvents were either employed as purchased or dried before use by standard laboratory procedures. Thin-layer chromatography (TLC) was carried out on 0.25 mm Leyan silica gel plates (60F-254). Column chromatography was performed on silica gel (200-300 mesh) as the stationary phase. $^1$H, $^{13}$C NMR, 2D NMR spectra were performed on Bruker Avance-500 NMR spectrometers. Chemical shifts are reported in ppm with residual solvents or tetramethylsilane (TMS) as the internal standards. The following abbreviations were used for signal multiplicities: s, singlet; d, doublet; dd, doublet of doublet; m, multiplet. Host-guest complexes were prepared by simply mixing the guests and hosts in 1: 1 stoichiometry in the corresponding solvent. Electrospray-ionization high-resolution mass spectrometry (ESI-HRMS) experiments were conducted on an applied Q-EXACTIVE mass spectrometry system. Circular Dichroism (CD) and UV-Vis spectra were recorded on an Applied Photo Physics Chirascan CD spectropolarimeter, using a 1 cm quartz cuvette. Fluorescent spectra were recorded on a spectro-fluorometer (Edinburgh FS5), using a 1 cm quartz cuvette. Specific rotations were measured on Rudolph Research Analytical Autopol I Polarimeter (589 nm) in a 1 dm length cell under 25 °C.

### Synthesis and characterization of chiral naphthotubes

The diisothiocyanate (561 mg, 0.92 mmol; in 60 mL $CH_2Cl_2$) and chiral diamine hydrochloride (Optical pure compounds with *R,R* / *S,S* configuration, 577 mg, 0.92 mmol; in 60 mL $CH_2Cl_2$) in two separate syringes were added dropwise via a double-channel syringe pump to the solution of N,N-Diisopropylethylamine (DIEA, 646 mg, 871 μL,

5.0 mmol) in $CH_2Cl_2$ (400 mL) during the course of 10 h. The resulting mixture was stirred at room temperature for 24 h. After the solvent was removed in vacuum, the residue was purified by column chromatography ($SiO_2$, $CH_2Cl_2$: MeOH = 1000/5) to give the compound **CT1** and **CT2** as a white solid. The enantiomers (*R,R* / *S,S*) are synthesized by similar methods. Specific rotation (*R,R*-**CT1**: $[\alpha]^{25}_D$ = +135.0 (*c*, 0.002, Dichloroethane), *R,R*-**CT2**: $[\alpha]^{25}_D$ = +10.0 (*c*, 0.002, Dichloroethane); *S,S*-**CT1**: $[\alpha]^{25}_D$ = −136.7 (*c*, 0.002, Dichloroethane), *S,S*-**CT2**: $[\alpha]^{25}_D$ = −10.0 (*c*, 0.002, Dichloroethane)). ESI-HRMS: m/z calculated for *R,R*-**CT1** $[M + H]^+$ $C_{70}H_{77}N_4O_8S_2$: 1165.5178, found 1165.5173 (error = −0.4 ppm); m/z calculated for *R,R*-**CT2** $[M + H]^+$ $C_{70}H_{77}N_4O_8S_2$: 1165.5178, found 1165.5178 (no error); m/z calculated for *S,S*-**CT1** $[M + H]^+$ $C_{70}H_{77}N_4O_8S_2$: 1165.5178, found 1165.5175 (error = −0.3 ppm); m/z calculated for *S,S*-**CT2** $[M + H]^+$ $C_{70}H_{77}N_4O_8S_2$: 1165.5178, found 1165.5173 (error = −0.4 ppm). The NMR and HPLC characterization data of the naphthotubes are provided in the Supplementary Information Figs. 10–13, 15–16 and Fig. 9.

### Determination of the association constants

The association constants for the complexes of **CT**s with most guests (except *S,S*-**CT2** with *S*-**5d**) are small ($<10^5\,M^{-1}$) and the chemical exchange is fast at the NMR timescale. Thus, we used direct NMR titrations to determine their association constants. The data were analyzed using the instrumental internal software package and fitted by "one set of binding sites" model to give the association constants ($K_a$). Non-linear fitting data are shown in Supplementary Figs. 44–107. For the cases (*S,S*-**CT2** complex with *S*-**5d**) with large association constants ($>10^5\,M^{-1}$) and fast exchange kinetics, the binding affinities were determined by competitive NMR titrations and using guest *S*-**5a** (binding constant with *S,S*-**CT2** is $1.2 \times 10^4\,M^{-1}$) as the outgoing guest. The data from competitive titrations was nonlinearly fitted[64] according to the equations developed by Prof. Werner Nau (available from their website, http://www.jacobs-university.de/ses/wnau). All $^1$H NMR titration experiments were repeated 3 times, and the averaged values and standard deviations are given.

### Data availability

The X-ray crystallographic coordinates for structures generated in this study have been deposited at the Cambridge Crystallographic Data Centre (CCDC), under deposition numbers 2268760 (for *R,R*-**CT2**·2Acetone), 2269592 (for *S,S*-**CT2**·2Acetone), 2247191 (for *R*-**5a**@*S,S*-**CT2**), 2247223 (for *S*-**5a**@*S,S*-**CT2**), 2247283 (for *S,S*-**5c**@*S,S*-**CT2**), 2247282 (for 2**Toluene**@*R,R*-**CT2**), 2269211 (for *S,S*-**5b**@*R,R*-**CT2**). These data can be obtained free of charge from the Cambridge Crystallographic Data Centre via www.ccdc.cam.ac.uk/data_request/cif. The authors declare that the data supporting the findings of this study are available within the article and its Supplementary Information Files. The atomic coordinates of structures for DFT calculation are provided as a Source Data file. Source data are provided with this paper.

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

## Acknowledgements

This research was financially supported by the National Natural Science Foundation of China (22125105, W.J.; 21801125, L.P.Y.; 22301127, L.H.), Department of Education of Guangdong Province (2020ZDZX2060, W.J.) and Guangdong Basic and Applied Basic Research Foundation (2022A1515110991, L.H.). The calculations were supported by Center for Computational Science and Engineering at Southern University of Science and Technology, and the CHEM high-performance supercomputer cluster (CHEM-HPC) located at department of chemistry, SUSTech.

## Author contributions

W.J. conceived and designed the experiments. S.-M.W. carried out the experiments and performed the DFT calculations. Y.-F.W., H.Y solved crystal structures. L.-P.Y., X.W., S.-M.W., Y.-F.W., L.H., L.S.Z., H.N., and Y.-T.Z. analyzed the data. L.-P.Y., X.W., S.-M.W., Y.-F.W. wrote the manuscript with contributions from all the authors.

## Competing interests

The authors declare no competing interests.
