## [Peer Review File · Nature Communications]

Chiral Recognition of Neutral Guests by Chiral Naphthotubes with a Bis-thiourea Endo-Functionalized CavityREVIEWER COMMENTS

Reviewer #1 (Remarks to the Author):

This work by Yang et al. describes two pairs of chiral naphthotubes containing a bis-thiourea endo-functionalized cavity, which showed highly enantiomerically selective recognition towards various neutral chiral molecules, particularly bioactive cyclic dipeptides, with an enantiomeric selectivity up to 17.0. Moreover, the mechanism of the chiral recognition has been revealed, which might be attributed to slight differences of multiple noncovalent interactions, as well as the orientation of binding and enthalpy-entropy compensation. Publication of the results is suggested after revisions noted as below.

1. For the chiral guests used for the chiral recognition, one pair of enantiomers instead of only one enantiomer (such as R-2, S-4a, and so on) will be more meaningful.
2. Compared with chiral guest 1, chiral guest 5b showed significant enhanced complexation ability and much highly enantioselective recognition. A reasonable explanation can be provided.
3. The absorption and emission changes of the naphthotubes in the presence of the chiral guests will be very helpful for mechanism investigation of the chiral recognition.
4. For the new chiral molecules, their optical rotations are suggested to be provided.

Reviewer #2 (Remarks to the Author):

This paper discusses the synthesis and binding properties of tube-like chiral hosts, which are able to enantiomerically discriminate in the binding of a series of chiral guest molecules. In terms of association constant value, enantioselectivities up to a factor of 17 are found. Some of the host-guest complexes are characterized by means of their solid state X-ray structures, in which it is clear that the host-guest binding is governed by multipoint interactions and differences in binding geometry. The binding mechanisms and the origins of the enantioselectivity are further discussed on the basis of spectroscopic measurements and DFT molecular modeling calculations.

The results in this paper are interesting and will appeal to the general supramolecular chemistry community. As the authors state, it is indeed far from straightforward to achieve

decent enantioselective host-guest binding in artificial systems. The authors have succeeded in designing a series of tube-like hosts that fulfill this desire, and have successfully applied the biologically-inspired concept of multipoint binding interactions to a synthetic system. Still, the newly designed receptors are elegantly simple, while at the same time they are able to bind significantly different types of guests. The new compounds have been characterized satisfactorily and the technical quality of the physical measurements is high. The precise binding mechanisms have been investigated in great detail and are convincing. It is my opinion that this work may become suitable for publication in this journal after the following comments have been addressed:

1. In the introduction, the authors mention previous research that reported other hosts that can achieve enantioselective binding. In the current work, a enantioselectivity factor of 17 is found, and it would be very informative if the authors would put this result in perspective to enantioselectivity factors found by the earlier reported systems (perhaps in the conclusion section).

2. When looking at the association constant values in Table 1, the paragraph about the preferences of the hosts for homochiral or heterochiral guest binding (lines 113-117) is somewhat confusing, in particular in terms of specification of the mentioned hosts that select heterochiral guests. For example, guest S-4a indeed preferentially binds in a heterochiral fashion to host RR-CT1, but in a homochiral fashion to the isomeric host SS-CT2. Same for guest R-4c. And guest S-4b binds with heterochiral preference to host RR-CT2, but homochiral to its isomer SS-CT1. A somewhat more exact description of the specific host-guest combinations would be desired here.

3. The statement in lines 147-151 in which the success or failure of crystallization of a host-guest complex is directly related to host-guest binding selectivity is much too strong (or even invalid), since crystallization may depend on many more factors than just host-guest binding strength. I thus suggest removal of this sentence.

4. Line 162: it is not clear what protons 5 and 5' are, where are they located in the molecules?

5. Lines 190 and 194: while the general readership of the journal might be aware of what DFT is, abbreviations like PCM and IGM are certainly not common knowledge and should be explained in much more detail.

6. The thermodynamic parameters as shown in Table S2 confuse me, as it seems that each of the diastereomeric host-guest complex pairs always have the same ΔG , ΔH and ΔS values. Should the numbers not represent the calculated *differences* in energy/enthalpy/entropy values between the diastereomeric partners, i.e. “ $\Delta\Delta$ ” values?

Reviewer #3 (Remarks to the Author):

Wang and co-workers reported the synthesis of two pairs of chiral naphthotubes via endo-functionalization strategy and their chiral recognition performance towards bioactive cyclic dipeptides. The mechanism of the chiral recognition was studied through various tests, as well as DFT calculations, and is attributed to multiple noncovalent interactions, shape matching, orientation, and enthalpy-entropy compensation. This work is interesting and the topic is important, yet, the overly rich influence factors still make the confuse to the design of host receptors for chiral recognition of bioactive molecules. Considering that the authors mainly analyzes the recognition behavior of substrate 5, it is suggested to further combine the chiral recognition mechanism of other substrates, especially substrate 4, to get more clear image about the main influencing factors. Therefore, I recommend reconsidering the manuscript after major revisions. The following issues also need revisions.

1.As shown in Fig.5, there is a difference in the CD spectrum before and after the guest visit, then similar to the 1H NMR spectrum, the CD spectrum should be provided for the titration experiment (J. Am. Chem. Soc. 2019, 141, 16382–16387) .

2.The $^1H,^1H$ -COSY NMR of different chiral conformational guest molecules coexisting with the host receptor should be added to the discussion to more clearly elucidate the differences in interactions.

3.In Fig.4, is these weak interactions such as hydrogen bond, $N-H\cdots\pi$ or $N-H\cdots\pi$ exist in the crystals will maintain when they dissolved in the solvents? Or greatly weakened? How about the effect of solvents on chiral recognition? In addition, I noticed that the squeeze in

PLATON are used to eliminate the distribution of solvent molecules, however, the influence from solvent molecules is considered in the discussion of the host-guest interactions in the crystals. Thus, I strongly suggest to determine the solvent molecules in these crystals without squeeze.

4. Fig. 5b, the color of R-5a@S,S-CT2 seems not match the label. Please check.

5. Fig. S67, the authors claims the peaks shift from 7.77 to 7.90 ppm with the concentration increasing, however, with the concentration increasing, the peaks at about 7.7 located at the right also enhanced, Why? Is these peaks shift to the right? Meanwhile, the spectra of 0.11 sample seems different from others.

6. For X-ray crystallography,

i) the GOOF, R1 and wR2 value of S,S-5c@S,S-CT2 are too high.

ii) The authors response all the Alert B is "The crystal has too weak diffraction to obtain high resolution data". I suggest that the authors put more effort into crystallographic testing to improve the quality of diffraction, for example by using MetalJet (Ga K α) or synchrotron radiation to obtain high quality data.

iii) In this manuscript, only the molecular structure unit diagrams are shown, and the stacking and interactions between molecules and molecules should also be represented by the stacking diagram.

Title: "Chiral Recognition of Neutral Guests by Chiral Naphthotubes with a Bis-thiourea Endo-Functionalized Cavity "

REVIEWER COMMENTS

Reviewer #1 (Remarks to the Author):

This work by Yang et al. describes two pairs of chiral naphthotubes containing a bis-thiourea endo-functionalized cavity, which showed highly enantiomerically selective recognition towards various neutral chiral molecules, particularly bioactive cyclic dipeptides, with an enantiomeric selectivity up to 17.0. Moreover, the mechanism of the chiral recognition has been revealed, which might be attributed to slight differences of multiple noncovalent interactions, as well as the orientation of binding and enthalpy-entropy compensation. Publication of the results is suggested after revisions noted as below.

Response: Thank you very much for the kind recommendation and the valuable suggestions! We have changed the manuscript accordingly.

1. For the chiral guests used for the chiral recognition, one pair of enantiomers instead of only one enantiomer (such as R-2, S-4a, and so on) will be more meaningful.

Response: Thanks for your helpful suggestions and we totally agree with you! However, only one enantiomer is commercially available for some chiral guests, especially for cyclic dipeptides (CDPs) guests, which are usually composed of L-amino acids (*Trends. Mol. Med.* **2014**, *20*, 551). We exactly have considered synthesizing these compounds ourselves, but it would require much synthetic effort (*Bull. Chem. Soc Jpn.* 1983, *56*, 568) for us, which is, however, not our focus in this research.

In fact, there are two methods for determining the enantioselectivities between a pair of enantiomeric guests and/or hosts. The first method involves utilizing one chiral host that binds to two enantiomeric guests, resulting in different association constants. The second method involves using two enantiomers of hosts, each with different association constants with the same chiral guest. The selectivity obtained from both methods should agree with each other quantitatively and qualitatively. Previous reports from our group and others have also verified the reliability of the two methods (*CCS Chem.* **2020**, *2*, 440; *Angew. Chem. Int. Ed.* **2022**, *61*, e202202527).

We obtained two pairs of enantiomers of hosts (including *syn*- and *anti*-configuration) through chiral synthesis. This allows us to determine the chiral selectivity by the second method, and the feasibility of this approach also has been verified by using chiral guests with two enantiomeric forms, such as the chiral guests 1 and 3.

2. Compared with chiral guest 1, chiral guest 5b showed significant enhanced complexation ability

and much highly enantioselective recognition. A reasonable explanation can be provided.

Response: Regarding the chiral guest *S,S*-**5b**, the amide bond, or peptide bond, is a more suitable hydrogen bond donor and acceptor due to the delocalized electron (Finkelstein, A. V., Ptitsyn, O. B., *Lecture 2. In Protein Physics (Second Edition)*, Academic Press: Amsterdam, **2016**; pp 17-25). More important, through the single crystal structure of the *S,S*-**5b**@*R,R*-**CT2**, it could be observed that NH- π interactions also play an important role in the host-guest binding. These factors ultimately contribute to the significant enhancement of complexation ability and higher enantioselectivity. Accordingly, we have included corresponding explanations in the revised manuscript.

3. The absorption and emission changes of the naphthotubes in the presence of the chiral guests will be very helpful for mechanism investigation of the chiral recognition.

Response: Thanks for your helpful suggestion. The spectra of absorption and emission changes of the naphthotubes in the presence of the chiral guests have been included in supporting information (Figs. S97-S102 and Figs. S111-S112)

4. For the new chiral molecules, their optical rotations are suggested to be provided.

Response: Thank you for the helpful suggestion. The optical rotations of six new chiral molecules have been obtained and provided in supporting information. Specific rotation (*R,R*-**6**: $[\alpha]_{\text{D}}^{25} = +0.9$ (*c*, 0.01, MeOH), *S,S*-**6**: $[\alpha]_{\text{D}}^{25} = -0.7$ (*c*, 0.01, MeOH), *R,R*-**CT1**: $[\alpha]_{\text{D}}^{25} = +135.0$ (*c*, 0.002, DCE), *R,R*-**CT2**: $[\alpha]_{\text{D}}^{25} = +10.0$ (*c*, 0.002, DCE), *S,S*-**CT1**: $[\alpha]_{\text{D}}^{25} = -136.7$ (*c*, 0.002, DCE), *S,S*-**CT2**: $[\alpha]_{\text{D}}^{25} = -10.0$ (*c*, 0.002, DCE).

Reviewer #2 (Remarks to the Author):

This paper discusses the synthesis and binding properties of tube-like chiral hosts, which are able to enantiomerically discriminate in the binding of a series of chiral guest molecules. In terms of association constant value, enantioselectivities up to a factor of 17 are found. Some of the host-guest complexes are characterized by means of their solid state X-ray structures, in which it is clear that the host-guest binding is governed by multipoint interactions and differences in binding geometry. The binding mechanisms and the origins of the enantioselectivity are further discussed on the basis of spectroscopic measurements and DFT molecular modeling calculations.

The results in this paper are interesting and will appeal to the general supramolecular chemistry community. As the authors state, it is indeed far from straightforward to achieve decent enantioselective host-guest binding in artificial systems. The authors have succeeded in designing a series of tube-like hosts that fulfill this desire, and have successfully applied the biologically-inspired concept of multipoint binding interactions to a synthetic system. Still, the newly designed receptors are elegantly simple, while at the same time they are able to bind significantly different types of guests. The new compounds have been characterized satisfactorily and the technical quality of the physical measurements is high. The precise binding mechanisms have been investigated in

great detail and are convincing. It is my opinion that this work may become suitable for publication in this journal after the following comments have been addressed:

Response: Thank you very much for the kind recommendation and the valuable suggestions! We have carefully addressed the identified issues and made extensive corrections accordingly.

1. In the introduction, the authors mention previous research that reported other hosts that can achieve enantioselective binding. In the current work, a enantioselectivity factor of 17 is found, and it would be very informative if the authors would put this result in perspective to enantioselectivity factors found by the earlier reported systems (perhaps in the conclusion section).

Response: Thank you for the kind suggestion, the corresponding information has been included in the manuscript.

2. When looking at the association constant values in Table 1, the paragraph about the preferences of the hosts for homochiral or heterochiral guest binding (lines 113-117) is somewhat confusing, in particular in terms of specification of the mentioned hosts that select heterochiral guests. For example, guest S-4a indeed preferentially binds in a heterochiral fashion to host RR-CT1, but in a homochiral fashion to the isomeric host SS-CT2. Same for guest R-4c. And guest S-4b binds with heterochiral preference to host RR-CT2, but homochiral to its isomer SS-CT1. A somewhat more exact description of the specific host-guest combinations would be desired here.

Response: Thank you for the kind suggestion. the *syn*-configured and *anti*-configured host exhibit different chiral preferences towards guests 3, 4a, 4b, and 4c. Furthermore, it can be observed that the shape of the guests and the variations in the substituent groups also influence the chiral preference. Based on the experimental results, we have marked with letter b and c in Table 1 to distinguish the homochiral and heterochiral preferences, and provided corresponding explanations in the revised manuscript to avoid misunderstanding.

3. The statement in lines 147-151 in which the success or failure of crystallization of a host-guest complex is directly related to host-guest binding selectivity is much too strong (or even invalid), since crystallization may depend on many more factors than just host-guest binding strength. I thus suggest removal of this sentence.

Response: Thank you for the helpful suggestion. We have removed this sentence accordingly.

4. Line 162: it is not clear what protons 5 and 5' are, where are they located in the molecules?

Response: Thanks for the kind suggestion. We have labeled the protons of the naphthotubes, including protons 5 and 5', in Figure 2a and Figs. S1-S4, and provided corresponding statement in the relevant sections of the maintext.

5. Lines 190 and 194: while the general readership of the journal might be aware of what DFT is, abbreviations like PCM and IGM are certainly not common knowledge and should be explained in

much more detail.

Response: Thank you for the kind reminder. We have provided explanations or replaced abbreviations with general terms, including but not limited to DFT, PCM, and IGM.

6. The thermodynamic parameters as shown in Table S2 confuse me, as it seems that each of the diastereomeric host-guest complex pairs always have the same delta-G, deltaH and deltaS values. Should the numbers not represent the calculated *differences* in energy/enthalpy/entropy values between the diastereomeric partners, i.e. “delta-delta” values?

Response: Thank you for the kind suggestion. We have redesigned Table S2 and added detailed explanations in supporting information.

Reviewer #3 (Remarks to the Author):

Wang and co-workers reported the synthesis of two pairs of chiral naphthotubes via endo-functionalization strategy and their chiral recognition performance towards bioactive cyclic dipeptides. The mechanism of the chiral recognition was studied through various tests, as well as DFT calculations, and is attributed to multiple noncovalent interactions, shape matching, orientation, and enthalpy-entropy compensation. This work is interesting and the topic is important, yet, the overly rich influence factors still make the confuse to the design of host receptors for chiral recognition of bioactive molecules. Considering that the authors mainly analyzes the recognition behavior of substrate 5, it is suggested to further combine the chiral recognition mechanism of other substrates, especially substrate 4, to get more clear image about the main influencing factors. Therefore, I recommend reconsidering the manuscript after major revisions. The following issues also need revisions.

Response: Thank you very much for the kind recommendation and the valuable suggestions! We have carefully addressed the identified issues and made extensive corrections accordingly.

Considering that the authors mainly analyzes the recognition behavior of substrate 5, it is suggested to further combine the chiral recognition mechanism of other substrates, especially substrate 4, to get more clear image about the main influencing factors.

Response: Thanks for your valuable suggestion. Based on the findings about chiral recognition of the guests 5, we proceeded to investigate the potential binding mode of chiral guests 4 in the cavity by theoretical calculation (Figs. S107-S110). The results revealed the substituents of guests and the shape of cavities both play crucial roles in constraining guest binding orientation. The paragraph about the investigation of the chiral recognition mechanism and corresponding calculated results have been added in the revised manuscript and supporting information.

1. As shown in Fig.5, there is a difference in the CD spectrum before and after the guest visit, then similar to the ¹H NMR spectrum, the CD spectrum should be provided for the titration experiment

(J. Am. Chem. Soc. 2019, 141, 16382-16387).

Response: Thank you for the kind suggestion, the CD titration experiments have been added in supporting information (Figs. S97-S102 and Figs. S111-S112).

2. The ¹H, ¹H-COSY NMR of different chiral conformational guest molecules coexisting with the host receptor should be added to the discussion to more clearly elucidate the differences in interactions.

Response: Thank you for the helpful suggestion. We have attempted to conduct ¹H, ¹H-ROESY experiments (as ¹H, ¹H-COSY NMR above) to obtain evidence of the differences in interactions between enantiomers (Figs. S91-S92). However, as the guest exchange is fast on the NMR time-scale, and thus the signals of the guest molecules were averaged and broadened, making it difficult to differentiate or detect them in the complex host signals. Although we were fortunate to observe a broad signal from a group of guests in the negative field, no NOE signals were detected between the chiral host and the guest, which could be attributed to signal averaging.

After careful analysis of data, we can obtain valuable evidence of the differences in interactions between enantiomers by utilizing the 1:1 host-guest NMR spectroscopic data as shown in Figs. S9-S18. When different enantiomers of the guest are bound to the same chiral host in a 1:1 ratio, or different enantiomers of the host are bound to the same chiral guest in a 1:1 ratio, different variations in chemical shifts were both observed, indicating potential differences in non-covalent interactions between the host and the guest. To validate that the differences in chemical shift changes were indeed due to variations in non-covalent interactions, we conducted 1:1 host-guest experiment using a strongly polar solvent, such as DMSO, to eliminate non-covalent interactions, including hydrogen bonds, NH- π and CH- π interactions. The experimental results demonstrated that different variations in chemical shifts both disappeared, providing evidence for the importance of non-covalent interaction in the chiral recognition in the liquid phase (Fig. S93).

3. In Fig.4, do these weak interactions such as hydrogen bond, N-H... π or N-H... π exist in the crystals will maintain when they dissolved in the solvents? Or greatly weakened? How about the effect of solvents on chiral recognition? In addition, I noticed that the squeeze in PLATON are used to eliminate the distribution of solvent molecules, however, the influence from solvent molecules is considered in the discussion of the host-guest interactions in the crystals. Thus, I strongly suggest to determine the solvent molecules in these crystals without squeeze.

Response: Thank you for the thoughtful suggestion. Generally, the solvent effect should have impact on the affinity of hosts and guests. On the one hand, the host-guest binding needs to overcome the solvation energy in solution, on the other hand, the solvent may also bind with the host to compete for the host-guest binding. An example of this phenomenon is the formation of a crystal of **2toluene@R,R-CT2** during the crystal growth process. As for the solvent effect on chiral recognition, we have proceeded the ¹H NMR titration experiments of **S,S-1@S,S-CT1** and **R,R-1@S,S-CT1** in CDCl₃. The affinity constants for **S,S-1@S,S-CT1** ($(1.4 \pm 0.1) \times 10^2$, $R^2 = 0.9999$)

and *R,R*-1@S,S-CT1 ($(2.9 \pm 0.2) \times 10^4$, $R^2 = 0.9996$) were obtained respectively. The enantioselectivity (4.9) obtained in CDCl₃ was consistent with that observed in toluene (4.6). The results revealed that the solvent effect almost has no influence on enantioselectivity, but it affects the affinity constants, which decreased.

Additionally, we performed theoretical calculations based on the single crystal structure and employed suitable solvent models (PCM and SMD solvent models) in the calculations. The theoretical calculations revealed that the energy differences associated with chiral selectivity aligned well with the energy differences observed in solution. Furthermore, the analysis of non-covalent interactions in the calculated host-guest structures demonstrated the presence of multiple non-covalent interactions between the hosts and guests.

As for X-ray crystallography, the single crystals were re-analyzed and refined without squeeze. In the crystal structures of S,S-5b@R,R-CT2 and S,S-5c@S,S-CT2, there are toluene molecules found within the interval of host-guest complexes. These toluene molecules could form CH- π interactions with the periphery of host to stabilize the crystal. However, the toluene molecules do not interact with the chiral guests within in hosts.

4. Fig. 5b, the color of R-5a@S,S-CT2 seems not match the label. Please check.

Response: Thanks for the kind reminder. The color of R-5a@S,S-CT2 in Fig. 5b has been changed accordingly to match the label.

5. Fig. S67, the authors claims the peaks shift from 7.77 to 7.90 ppm with the concentration increasing, however, with the concentration increasing, the peaks at about 7.7 located at the right also enhanced, Why? Is these peaks shift to the right? Meanwhile, the spectra of 0.11 sample seems different from others.

Response: Thank you for the kind suggestion. In the titration, as the concentration increases, the peaks at about 7.79 located at the right would shift downfield, and the peaks at about 7.83 located at the left would shift upfield. In the spectra of 0.11 sample, two peaks crossed and fused together to form a set of peaks. To improve clarity and avoid confusion, we added markers to Figure S69 (previously referred to as Figure S67).

6. For X-ray crystallography,

i) the GOOF, R1 and wR2 value of S,S-5c@S,S-CT2 are too high.

Response: Thank you for the kind suggestion. We have conducted a new growth and analyzed the single crystal of S,S-5c@S,S-CT2 and made the parameters reasonable. And relevant data has been updated in table S1.

ii) The authors response all the Alert B is "The crystal has too weak diffraction to obtain high resolution data". I suggest that the authors put more effort into crystallographic testing to improve the quality of diffraction, for example by using MetalJet (Ga K α) or synchrotron radiation to obtain

high quality data.

Response: Thank you for your kind suggestion. In order to enhance the quality and reliability of the single crystal data, we have conducted a new growth and testing process for the single crystals with low-quality diffraction. All single crystal X-ray data were collected using a Bruker D8 VENTURE instrument with Ga K α radiation. And relevant data has been updated in table S1.

iii) In this manuscript, only the molecular structure unit diagrams are shown, and the stacking and interactions between molecules and molecules should also be represented by the stacking diagram.

Response: Thanks for your helpful suggestion. We have provided stacking diagrams of all single crystals viewed along a, b and c-axis. The diagrams illustrated the presence of CH- π (ArH- π) interactions among hosts and/or hosts and toluene molecules in the crystal. These interactions play a crucial role in achieving crystal stability.

REVIEWERS' COMMENTS

Reviewer #1 (Remarks to the Author):

The authors have revised the manuscript carefully according to the comments and suggestions of the reviewers, so publication is now recommended.

Reviewer #2 (Remarks to the Author):

The authors have addressed all my comments satisfactorily and I can now recommend publication of this work

Reviewer #3 (Remarks to the Author):

It is clear from their response that Yang and coworkers have carefully considered the reviewers' comments, which generally pertained to the below key points:

Missing supporting information

The revisions, which are thoroughly accounted for in the response document, have addressed these key points. In particular, the points I raised in my initial review relating to the non-covalent interactions and solvent influence have been resolved.

Following these changes manuscript could be published as it, subject to one minor change detailed below:

#Minor

Please provide files as .xyz. related to the optimized atomic coordinates of the structure included in the DFT calculation. Only this way can this data be straightforwardly used by the community.

Title: "Chiral Recognition of Neutral Guests by Chiral Naphthotubes with a Bis-thiourea Endo-Functionalized Cavity "

REVIEWER COMMENTS

Reviewer #1 (Remarks to the Author):

The authors have revised the manuscript carefully according to the comments and suggestions of the reviewers, so publication is now recommended.

Response: Thank you very much for your recommendation to our work and the valuable suggestions!

Reviewer #2 (Remarks to the Author):

The authors have addressed all my comments satisfactorily and I can now recommend publication of this work

Response: Thank you very much for your affirmation to our work and the valuable suggestions!

Reviewer #3 (Remarks to the Author):

It is clear from their response that Yang and coworkers have carefully considered the reviewers' comments, which generally pertained to the below key points:

Missing supporting information

The revisions, which are thoroughly accounted for in the response document, have addressed these key points. In particular, the points I raised in my initial review relating to the non-covalent interactions and solvent influence have been resolved.

Following these changes manuscript could be published as it, subject to one minor change detailed below:

#Minor

Please provide files as .xyz. related to the optimized atomic coordinates of the structure included in the DFT calculation. Only this way can this data be straightforwardly used by the community.

Response: Thank you very much for the kind recommendations and constructive suggestions! We have provided the .xyz files of atomic coordinates of the structure.